# Embedded timing and alert device triggered by total dissolved solids (TDS) for monitoring disinfection duration in acidic electrolyzed oxidizing water

Wei Zheng[1], Hongxia Xu[2]*, Yuan Mao[3], Fucheng He[2], Ping Gui[1]

1 Central Sterile Supply Department, Sichuan Clinical Research Center for Cancer, Sichuan Cancer Hospital & Institute, Sichuan Cancer Center, Affiliated Cancer Hospital of the University of Electronic Science and Technology of China, Chengdu, China, 2 Central Sterile Supply Department, Fengdu General Hospital, Chongqing, China, 3 Central Sterile Supply Department, No. 363 Hospital, Chengdu, China

* cqxuting@163.com

## Abstract

To address the absence of a reliable timing and alerting tool for acidic electrolyzed oxidizing water (AEOW) disinfection of medical instruments in the Central Sterile Supply Department (CSSD), we developed an embedded timing and alert device triggered by total dissolved solids (TDS) sensing. Built around an Arduino development board, the device integrated a TDS sensor module, a rechargeable battery, and a wireless charging coil. Encapsulated with potting adhesive, it achieved an IP67 protection rating. It can identify the AEOW environment through TDS value detection to automatically start/stop timing, and is equipped with features including light alerts, data storage, and traceability. Medical instruments disinfected with AEOW in the CSSD of a hospital were selected as the test objects. Digital kitchen timers were used as the control group in September 2025, and TDS-triggered timers were adopted as the experimental group in October 2025. A comparative analysis was conducted on the disinfection timing execution rate and disinfection duration compliance rate between the two groups. The results demonstrated that the disinfection timing execution rate of 94.92% (655/690) and the disinfection duration compliance rate of 98.26% (678/690) in the experimental group were higher than those of 84.88% (713/840) and 93.69% (787/840) in the control group, with statistically significant differences (P < 0.001). Characterized by low cost and simple operation, the TDS-triggered timer provides a reliable, automated, and traceable timing solution for AEOW disinfection in CSSD. It also facilitates digital management of the disinfection process, and is expected to improve the quality of AEOW disinfection and enhance patient safety.

**Data availability statement:** All relevant data are within the paper and its Supporting Information files.

**Funding:** The author(s) received no specific funding for this work.

**Competing interests:** The authors have declared that no competing interests exist.

## Introduction

The Central Sterile Supply Department (CSSD) is a critical unit within healthcare facilities, responsible for the reprocessing of reusable medical instruments [1,2]. As a core department for infection prevention, CSSD ensures that all post-use instruments are properly processed to prevent nosocomial infections and safeguard patient safety [3,4]. The reprocessing workflow for medical instruments mainly includes three steps: cleaning, disinfection, and sterilization. Disinfection is essential to reduce the microbial load on medical instruments and prepare them for the final sterilization step; inadequate disinfection may result in sterilization failure [5–8]. Disinfection methods are primarily categorized into physical and chemical approaches. Dry heat, moist heat, radiation, and filtration belong to physical disinfection methods, among which moist heat disinfection is the simplest to operate and the most cost-effective, making it the most widely used in CSSD [9–11]. However, moist heat disinfection is only suitable for moisture- and heat-resistant medical instruments [7]. Other instruments require chemical disinfection methods, which include alcohol, glutaraldehyde, chlorine-based disinfectants, peracetic acid, and acidic electrolyzed oxidizing water (AEOW) [12–16].

AEOW is a colorless and transparent solution produced by the electrolysis of aqueous sodium chloride solution. It features a high oxidation-reduction potential (≥ 1100 mV), a low pH value (2.0–3.0), and contains hypochlorous acid, and is classified as a high-level disinfectant [5,17]. AEOW exhibits a broad antimicrobial spectrum, offers convenient operation with no on-site preparation required and a short disinfection time, and is environmentally friendly with minimal irritation to the skin and mucous membranes of technicians and no chemical residue. It is thus widely applied in the food processing, animal husbandry, and healthcare fields [17–25]. The operating procedure for AEOW disinfection is to immerse cleaned medical instruments in AEOW for two minutes, then remove and drain them [5,26]. The disinfection efficacy of AEOW is highly time-dependent; two minutes can achieve a five-log reduction of *Mycobacterium tuberculosis, Escherichia coli, Enterococcus faecalis, Pseudomonas aeruginosa, Bacillus subtilis var. niger spores, methicillin-resistant Staphylococcus aureus,* and *Candida albicans*. If the immersion time is less than two minutes, microbial residues may remain, and this defect lacks visually identifiable indicators, thereby posing potential risks to subsequent packaging and sterilization. Conversely, no negative effects have been observed when the immersion time is extended to more than 2 minutes [27].

When instrument sets (a batch of medical instruments held in wire mesh trays) are disinfected in AEOW, there is no visible change to the AEOW, medical instruments, or wire mesh trays. Additionally, technicians often concurrently handle multiple decontamination tasks and cannot continuously monitor the disinfection process of individual wire mesh trays. Therefore, the common challenge when using AEOW for disinfection is accurately recording the immersion time of instrument sets. Currently, the prevalent method in CSSD is to use digital kitchen timers for AEOW disinfection timing due to their simplicity of operation, low cost (less than $1), and alert function [28]. The digital kitchen timers are fixed on the wall beside the AEOW sink and set for

a two-minute countdown. Technicians immerse the instrument sets in AEOW and press the start button to initiate timing. After two minutes, the timer emits a continuous beeping sound to remind technicians to remove the instruments. However, this method of using digital kitchen timers has several obvious drawbacks. First, the timers lack a data storage function and cannot trace the time of each disinfection, even the most recent cycle. Second, a single timer can only record one time point; when multiple wire mesh trays with medical instruments are immersed in AEOW at different times, the timer cannot match and time each wire mesh tray individually. Third, the alert function has a design flaw: the beeping sound automatically stops after one minute, and the screen displays the original two-minute countdown value. If technicians fail to perceive the alert signal in a timely manner, they may be misled into thinking the timer was never activated and restart it (a common occurrence).

Besides using digital kitchen timers, there is currently no better method to accurately record the AEOW disinfection duration of medical instruments. Therefore, we designed a timing device suitable for AEOW disinfection scenarios based on embedded technology, which features automatic start/stop, light alerts, data storage, and waterproof operation. Total dissolved solids (TDS) refers to the total amount of various inorganic and organic substances dissolved in water, which can be represented by electrical conductivity with a positive correlation between the two [29,30]. Preliminary experiments have revealed that the electrical conductivity of AEOW is significantly higher than that of tap water and purified water. Taking this characteristic into account, we selected TDS as the activation signal for the timing device and set the activation threshold between that of AEOW and tap water to prevent false activation of the device in tap water and purified water.

## Materials and methods

### Device design and fabrication

The TDS-triggered timer (125 mm × 40 mm × 25 mm, 165 g) was built around an Arduino development board and consists of an integrated microcontroller module, a TDS sensor module, a rechargeable battery with a wireless charging coil, a waterproof button, LED lights, and a 3D-printed casing (Fig 1). A custom printed circuit board (PCB) was designed to integrate all hardware modules, improving circuit stability and overall integration (Fig 2). The assembled electronic components were finally potted in silicone encapsulant.

*Integrated Microcontroller Module:* The system takes the ESP-WROOM-32E-N16 wireless module as the control core. The module integrates an ESP32 microcontroller (dual-core CPU, Wi-Fi/Bluetooth) and 16 MB of flash memory, handling data processing and wireless communication.

*TDS Sensing Module:* It comprises a signal adapter board (42 mm × 32 mm, 3.3–5.5 V input, 0–2.3 V analog voltage and high/low level (1/0) output, 3–6 mA operating current) and a TDS probe with two 5-mm pins.

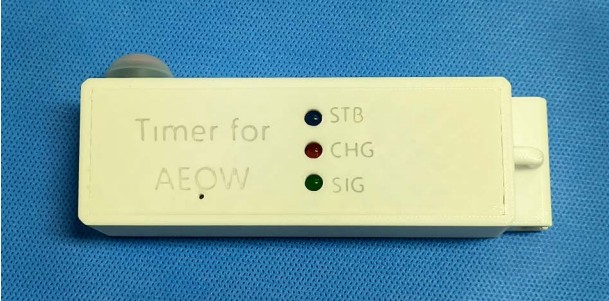

**Fig 1. Physical image of the TDS-triggered timer.**

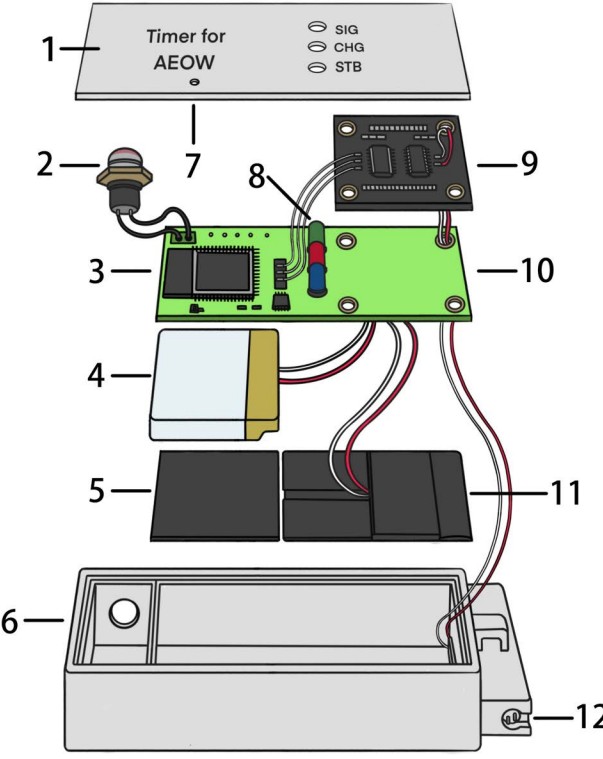

**Fig 2. Structural schematic diagram of the TDS-triggered timer.** 1. Top cover; 2. Waterproof switch; 3. Integrated microcontroller module; 4. Rechargeable battery; 5. Magnetic shielding sheet; 6. Casing; 7. Reset hole; 8. LEDs; 9. TDS module; 10. PCB board;11. Wireless charging coil; 12. TDS probe.

*Battery and Wireless Charging Module*: A lithium-ion battery (51 mm × 34 mm × 6 mm, 1200 mAh, 3.7 V output) and a wireless charging module (100–200 kHz operating frequency, rectified and regulated to 5 V DC, maximum 1 A output, 5 W power).

## Functions and operation

The finished device features such key functions as waterproofing, automatic start/stop, timing alerts, wireless charging, data storage, and wireless data transmission. By encapsulating all components with potting adhesive, the device was rated at IP67 for water and dust protection. Practical tests demonstrated that the device could still work normally after being continuously immersed in AEOW for 7 days (repeated 5 times). The TDS sensing module incorporates a voltage divider resistor structure and hardware binary decision logic. The output analog voltage amplitude is determined by the conductivity of the measured medium, with a positive correlation between the two (higher medium conductivity yields higher output analog voltage). The hardware threshold is implemented through a 1 V hardware comparator, against which the output analog voltage is compared. When the probe is immersed in AEOW, the high conductivity of AEOW results in an output voltage exceeding 1 V, causing the TDS sensing module to output a high level (1) that wakes the ESP32 microcontroller to initiate timing. When the probe is in tap water, purified water, or air, the low conductivity of the medium results in an output voltage below 1 V, and the sensing module outputs a low level (0), maintaining the device in sleep mode.

The new device is equipped with a green LED that flashes in a double-blinking pattern during timing. After 2 minutes, it remains steadily lit to indicate that the AEOW disinfection time is sufficient and the medical instruments can be removed.

Additionally, it incorporates a red LED for charging indication and a blue LED for charging complete status. The device is also furnished with a function button: a short press initiates battery level detection, with the green LED flashing three times to signal sufficient battery power. On-board memory can store up to 200 records, each containing the duration and end time of an instrument set disinfection cycle; when the memory reached full capacity, the oldest record was overwritten automatically. The device was required to be used within local Wi-Fi coverage to synchronize the real-time clock and ensure complete data entries (otherwise only the disinfection duration was logged). A 10-second long-press forces the green LED to blink continuously, entering network-configuration mode, from which a computer can read, export, or erase all stored records (Fig 3). The device exits configuration mode and enters sleep after five minutes if no action is taken; during AEOW disinfection timing, the function button was disabled to prevent accidental operation.

At the start of each workday, CSSD technicians checked the battery level of the device and charged it if depleted. During manual cleaning, when the AEOW disinfection step was reached, the technician placed the device LED-side up in a wire mesh tray that held instrument sets, then immersed the tray together with the device, which was required to be fully submerged. The wire mesh tray and device were removed together when the green LED remained steady, thus completing the AEOW disinfection process. If multiple wire mesh trays were processed, each tray was required to be equipped with its own TDS-triggered timer to ensure one-to-one correspondence and avoid timing confusion. Finally, the timer was removed from the tray and placed in a designated area for later use. Every Monday all stored data were exported and reviewed, with particular attention to cycles shorter than 2 minutes. If such cases were identified, surveillance footage was reviewed to investigate the circumstances, determine the root cause, and implement corrective actions.

## Observation index

To evaluate the practical effectiveness of the new device, we compared digital kitchen timers (control group) with the TDS-triggered timers (experimental group). Testing was carried out in a hospital CSSD where approximately 100 surgical procedures were performed daily. Each surgical procedure used 1–3 instrument sets, 20–50 of which required AEOW disinfection. The test period was from September to October 2025: digital kitchen timers were used in September, and the

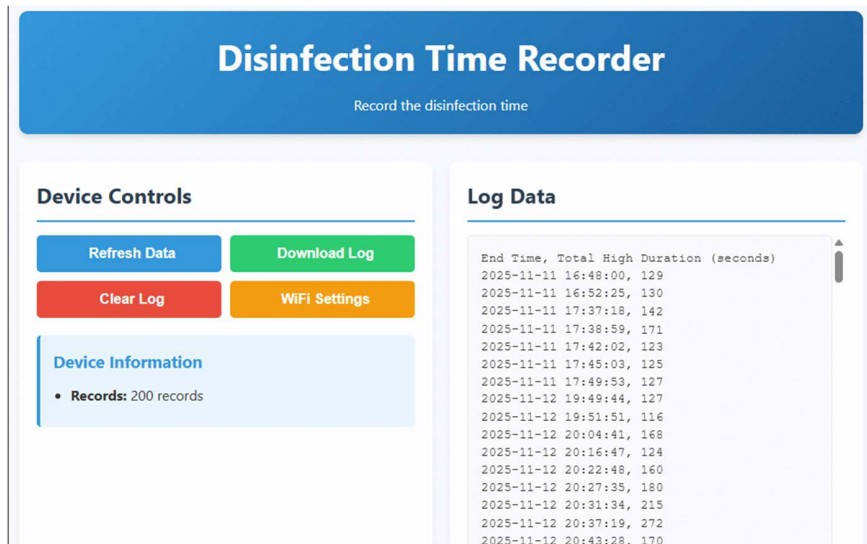

**Fig 3. Web interface for data review.**

new devices were adopted in October. During the test period, the technicians and workflow remained unchanged, with surgical workload fluctuating within the normal range.

In the control group, after immersing the instrument sets in AEOW, technicians pressed the start button of the wall-mounted timer to initiate the countdown (pre-set to two minutes). After the countdown ended, the timer emitted a continuous beeping sound, and technicians took out the instrument sets upon hearing the alert. In the experimental group, technicians placed one device in each wire mesh tray; the device's green LED blinked intermittently during immersion and became steadily lit after 2 minutes, prompting removal. The main observation indicators were the disinfection timing execution rate and the disinfection duration compliance rate, with data collected by reviewing video surveillance recordings. *Disinfection timing execution rate:* The proportion of instrument sets for which timing was performed among all instrument sets requiring AEOW disinfection. In the control group, timing was deemed initiated when technicians pressed the start button of the digital kitchen timer after immersing the instrument set in AEOW; in the experimental group, it was deemed initiated when technicians immersed the TDS-triggered timer together with the instrument set in AEOW. Failure to perform the above operations was recorded as not initiated.

*Disinfection duration compliance rate:* The proportion of instrument sets with an actual disinfection duration of at least 2 minutes during AEOW disinfection relative to the total number of instrument sets requiring AEOW disinfection, where the actual disinfection duration was defined as the time interval from the moment the entire instrument set was completely submerged below the AEOW surface (start time) to the moment it was removed from the AEOW (end time).

We did not compare operation time and data traceability. Regarding operational time, the digital kitchen timer required 1–2 button presses, whereas the TDS-triggered timer required placement into and removal from the wire mesh tray. Both methods took a very short time (less than three seconds), so any difference was negligible. In terms of data traceability, the digital kitchen timer is not equipped with any data recording or storage functionality, precluding the traceability of disinfection process data. In contrast, the TDS-triggered timer enables the traceability of disinfection duration via its built-in storage and wireless export functions. This fundamental difference in traceability capability between the two tools obviates the need for statistical comparison in the present study.

## Statistical analysis

Data were analyzed with R 4.5.1. Count data were described as frequencies and percentages, and inter-group comparisons used the test. A P-value $<0.05$ was considered statistically significant.

## Results

### Comparison of disinfection timing execution rates between two groups

The control and experimental groups included 840 and 690 instrument sets requiring AEOW disinfection, respectively. The disinfection timing execution rate of the experimental group was 94.92% (655/690), which was significantly higher than that of the control group (84.88%, 713/840). The difference was statistically significant, = 39.33, P<0.001 (Table 1).

Table 1. Comparison of disinfection timing execution rates between two groups (n).

| Groups | Performed disinfection timing | Did not perform disinfection timing |
|---|---|---|
| Control group (n = 840) | 713 | 127 |
| Experimental group (n = 690) | 655 | 35 |
| value | 39.33 | |
| P-value | < 0.001 | |

## Comparison of disinfection duration compliance rates between two groups

The disinfection duration compliance rate of the experimental group was 98.26% (678/690), which was significantly higher than that of the control group (93.69%, 787/840). The difference was statistically significant, = 18.35, P<0.001 (Table 2).

## Discussion

The research results showed that compared with digital kitchen timers, the use of the TDS-triggered timer increased the disinfection timing execution rate from 84.88% to 94.92% and the disinfection duration compliance rate from 93.69% to 98.26%; both differences were statistically significant (P<0.001). This improvement ensures more effective disinfection of medical instruments and reduces the potential risk of infection caused by inadequate disinfection. The improvement in both disinfection timing execution and disinfection duration compliance rates can also be explained by the Hawthorne effect, which refers to the phenomenon in which study participants unconsciously alter their behavior when aware of being observed [31–33]. The TDS-triggered timer's data storage and traceability functions make technicians aware that managers can query all usage records (frequency and duration of each disinfection), prompting them to use the new device correctly and proactively during disinfection. This behavioral factor amplifies the benefits of the technical improvement. Notably, 65 instrument sets across both groups failed to meet the required disinfection time, with durations ranging from 55–113 seconds. However, because all these sets subsequently underwent sterilization [5], the final state remained sterile and no substantive impact on medical safety occurred [34,35]. In addition, targeted retraining was provided to technicians who failed to initiate disinfection timing or achieved non-compliant disinfection durations to standardize their operational practices.

A single digital kitchen timer can record only one interval. When multiple wire mesh trays were immersed in AEOW at short intervals, the timer could not time each tray separately. If timing was initiated when the first tray was immersed and all trays were removed simultaneously upon countdown completion, the instruments placed later had insufficient disinfection time (i.e., early timing initiation and delayed immersion). Even preparing multiple timers does not solve this problem effectively, because wire mesh trays and instrument sets were highly similar, and technicians often handled multiple cleaning tasks simultaneously, easily confusing the match between timers and trays, which also resulted in insufficient disinfection time for some medical instruments. In contrast, the TDS-triggered timer was placed directly inside each wire mesh tray, achieving physical association and ensuring one device per instrument set. This approach avoided multitasking interference and reduced the recording confusion inherent in digital kitchen timer use. Moreover, the new device only started timing after being fully submerged in AEOW, ensuring that the start point coincided with actual disinfection onset and preventing false compliance caused by premature initiation of digital kitchen timers by technicians.

The new device was extremely user-friendly with a low learning curve. Technicians only needed to master battery charging, battery level checking, and the indicator meanings of the 3 LEDs to operate the device proficiently, even novice technicians could master it quickly. Digital kitchen timers required multiple manual start and stop operations, and initiation of timing occurred frequently. In contrast, the TDS-triggered timer automatically detected AEOW and initiated timing, eliminating missed timing initiation and improving process automation without increasing the workload

**Table 2. Comparison of disinfection duration compliance rates between two groups (n).**

| Groups | AEOW disinfection duration compliant | AEOW disinfection duration non-compliant |
|---|---|---|
| Control group (n=840) | 787 | 53 |
| Experimental group (n=690) | 678 | 12 |
| value | 18.35 | |
| P-value | < 0.001 | |

of technicians. In addition, the digital kitchen timer's alert sound stopped automatically after one minute and the screen reverted to the preset 2-minute countdown, which could mislead technicians into believing that disinfection timing had never been initiated. The green LED on the TDS-triggered timer remained steadily lit after completing the two-minute timing, ensuring that technicians correctly received the disinfection completion signal and reducing rework caused by misjudgment.

The core innovation of the new device lies in its TDS-based liquid identification method, which accurately distinguish between purified water, tap water, and AEOW. The conductivity threshold for device activation was determined to be 1176±47.81 μS/cm in laboratory tests, and the TDS-triggered timer only initiated timing when the solution conductivity exceeded this threshold. In this study, 5 independent samples were collected for tap water, purified water, and AEOW respectively, with 3 replicate conductivity measurements per sample, resulting in a total of 45 valid data points. The results demonstrated that the conductivity of tap water was 389.93±37.56 μS/cm, purified water was 2.97±0.76 μS/cm, and AEOW was 2343.13±103.03 μS/cm. During AEOW generation, NaCl dissociates into $Na^+$ and $Cl^-$ ions, and a large number of $H^+$ ions are produced at the anode, resulting in the extremely high conductivity of AEOW [19,20,25,26]. It was evident that the electrical conductivity values of both tap water and purified water were substantially lower than the device activation threshold, ensuring no false positive or false negative readings in practical applications. This study also tested 3 types of multi-enzymatic cleaning agents and 2 types of alkaline cleaning solutions used in CSSD (including products from China and the United States). At the standard dilution concentrations recommended by the manufacturers' instructions, the conductivity of all cleaning agents remained significantly below the device activation threshold, preventing inadvertent device triggering. Under three experimental conditions—4 hours of continuous AEOW usage, 24 hours of indoor standing, and 1:1 dilution with purified water—the reduction in electrical conductivity was minimal, and the device could still be activated stably. The new device initiates the timing function only when an AEOW environment is detected, which means it can verify whether the liquid used for disinfection is indeed AEOW and prevent the extreme scenario where purified water or tap water is mistakenly added to AEOW sinks and used for disinfection purposes. The new device did not accumulate timing time; once removed from AEOW, timing stopped immediately, and re-immersion started timing from zero. This design effectively resolved timing confusion when wire mesh trays were repeatedly immersed in and removed from AEOW, ensuring that each disinfection process was recorded independently. Cost-effectiveness was a key consideration in the device design, with the material cost per unit being approximately $10 and no consumables required for its use. In contrast, although digital kitchen timers had a low unit price, they were prone to damage and required frequent battery replacement, resulting in high long-term usage costs.

The TDS-triggered timer is not merely a timing tool but also an entry point for digital management of disinfection workflows. Its built-in data storage and wireless transmission functions made the start time and duration of each disinfection process traceable, thereby providing a foundation for quality monitoring, accountability, and process optimization [36,37]. Digital management also helped identify weak points in the disinfection process, such as inadequate disinfection times during certain periods or by certain technicians, enabling targeted scheduling adjustments or training. Furthermore, application of the new device produced a positive behavioral impact: technicians were more inclined to comply with protocols when aware that their actions were recorded, which invisibly improved overall operational quality [38].

In this study, retrospective review of video surveillance recordings was adopted as the sole data source, and the data stored by the TDS-triggered timers were not used for statistical analysis. The reasons for this design are twofold. First, the digital kitchen timers used in the control group lacked any data recording and storage capabilities, meaning all disinfection-related data for this group could only be obtained via retrospective review of video surveillance records. If device-stored data had been used for the experimental group, divergent data sources would have introduced detection bias, resulting in a loss of comparability between the two groups' outcomes. A unified data collection method was therefore critical to ensuring the validity of intergroup comparisons. Second, the operational workflow of the TDS-triggered timer involved a specific sequence: upon submersion in AEOW, a change in TDS conductivity activated the

CPU, followed by network connection for real-time clock synchronization, after which formal timing was initiated. This workflow incurred a systematic delay of 2.75±0.07 seconds, resulting in the device-stored timing duration being slightly shorter than the actual AEOW immersion duration of the instrument sets. Using device-stored data as the evaluation criterion would have introduced bias into the assessment of the disinfection duration compliance rate for the experimental group, whereas the actual immersion duration recorded via video surveillance more objectively reflected the real clinical disinfection operational status. The systematic delay of 2.75±0.07 seconds during the operation of TDS-triggered timer was unavoidable. However, additional immersion of the instrument sets in AEOW for a few seconds would neither reduce the disinfection efficacy nor cause any adverse effects on the material and functional properties of the medical instruments. Therefore, this short systematic delay had no significant impact on the core conclusions of the present study or its clinical application.

### Limitations and future work

This study had a notable methodological limitation: the use of a non-randomized sequential trial design, in which September and October 2025 were designated as the control and experimental phases, respectively, which introduced the potential for confounding due to temporal effects. Such bias was minimized through rigorous study controls: the same participating technicians were involved, the workflow remained unchanged, and the number of instrument sets requiring AEOW disinfection was stable throughout the study period, with no adjustments to management policies or work schedules. Additionally, technicians were fully aware that their behaviors were under video surveillance throughout the study, resulting in a consistent Hawthorne effect on both groups. While a randomized crossover trial design has theoretical advantages, it has significant practical limitations in CSSD settings: grouping by sink or team would introduce confounding due to differences in technicians' operational habits and uneven allocation of instrument sets, while random switching between the two timing methods at the same sink would increase the workload and the risk of operational errors. For these reasons, this design was not adopted in the present study. Furthermore, the primary outcome measures of this study were objective behavioral indicators, and the core difference between digital kitchen timers and TDS-triggered timers (i.e., the presence or absence of data traceability) imposed a definitive causal behavioral constraint on technicians. This core mechanism is independent of the study design format, and thus the non-randomized sequential design did not affect the reliability of the core conclusions. This study was conducted solely in the CSSD of a single hospital, which limits the representativeness of the findings. Future research should be carried out in multicenter settings with large sample sizes to further verify the generalizability of the conclusions drawn from this study.

Future work will focus on optimizing the TDS-triggered timer for AEOW disinfection across multiple hardware and software dimensions to better adapt to real-world CSSD scenarios. First, the current device relies on a single green LED to indicate both the ongoing and completion statuses of AEOW disinfection, resulting in weak light signals and limited visibility. We plan to increase the number of green LEDs and optimize their layout to enhance signal visibility, ensuring technicians can accurately track the disinfection progress of each wire mesh tray even when handling multiple trays simultaneously. Second, regarding power consumption and battery life, we will reduce standby power draw through software optimization (adjusting power management algorithms) and hardware upgrades (adopting ultra-low-power components and increasing battery capacity), thereby extending battery life and reducing the burden of frequent charging. Third, we will enrich the data interaction and monitoring functions by expanding data storage capacity, developing a dedicated mobile application to enable real-time automatic upload of disinfection data (thus reducing the workload of manual data export), and implementing an alarm mechanism. When the detected disinfection duration is less than two minutes, instant notifications will be pushed to the mobile device, allowing managers to intervene promptly. Statistical analysis functions will also be added to support visual presentation and trend analysis of key metrics such as disinfection timing execution rate and disinfection duration compliance rate. Fourth, we will minimize the device size by simplifying the PCB layout, selecting more compact components, and optimizing the 3D-printed casing structure. This will reduce the volume while

maintaining the IP67 protection rating, further enhancing the device's suitability and convenience in AEOW disinfection scenarios.

During the experiment, we simultaneously measured the electrical conductivity of three disinfectants: 2000 mg/L chlorine-containing disinfectant solution, a 1:75 dilution of binary peracetic acid disinfectant, and 2% glutaraldehyde disinfectant solution. The results showed that the electrical conductivity of all three exceeded the activation threshold of the TDS-triggered timer, which provided a feasible basis for extending the device's timing application to the aforementioned disinfectant solutions. In-depth research can be conducted on this direction in subsequent studies. Additionally, we can draw on the RFID technology adopted in the timing of flexible endoscope disinfection to further improve the full-cycle management of the disinfection process and data traceability.

## Conclusion

The TDS-triggered timer represents an innovative research achievement tailored for AEOW disinfection of medical instruments in CSSD. It integrates core functions including waterproof performance, automatic start and stop, timing alerts, wireless charging, data storage, wireless data transmission, and data traceability. In practical applications, the device has significantly improved the disinfection timing execution rate and duration compliance rate, standardized operational procedures, and is expected to become an important tool for the standardized management of immersion disinfection for medical instruments in CSSD. It provides robust support for ensuring the disinfection quality of medical instruments and reducing the risk of healthcare-associated infections.

## Supporting information

**S1 File. Research data.**
(XLSX)

**S2 File. Software code.**
(DOCX)

**S3 File. List of equipment and consumables for the experiment.**
(XLSX)

## Author contributions

Conceptualization: Wei Zheng, Hongxia Xu, Yuan Mao, Fucheng He.

Data curation: Wei Zheng, Yuan Mao, Fucheng He, Ping Gui.

Formal analysis: Hongxia Xu, Ping Gui.

Investigation: Fucheng He, Ping Gui.

Methodology: Yuan Mao, Fucheng He, Ping Gui.

Project administration: Wei Zheng, Hongxia Xu.

Software: Wei Zheng, Yuan Mao, Fucheng He, Ping Gui.

Supervision: Hongxia Xu, Yuan Mao.

Validation: Hongxia Xu, Fucheng He.

Visualization: Ping Gui.

Writing – original draft: Wei Zheng.

Writing – review & editing: Wei Zheng, Hongxia Xu, Yuan Mao, Fucheng He, Ping Gui.

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
