## [Decision Letter · Decision Letter 0]

9 Feb 2026

PONE-D-25-61065Embedded timing and alert device triggered by total dissolved solids (TDS) for monitoring disinfection duration in acidic electrolyzed oxidizing waterPLOS One

Dear Dr. Hongxia Xu,

Thank you for submitting your manuscript to PLOS ONE. After careful consideration, we feel that it has merit but does not fully meet PLOS ONE’s publication criteria as it currently stands. Therefore, we invite you to submit a revised version of the manuscript that addresses the points raised during the review process.

We look forward to receiving your revised manuscript.

Kind regards,

Andrey Nagdalian

Academic Editor

PLOS One

Journal Requirements:

Reviewers' comments:

Reviewer's Responses to Questions

**Comments to the Author**

1. Is the manuscript technically sound, and do the data support the conclusions?

Reviewer #1: Partly

Reviewer #2: Yes

2. Has the statistical analysis been performed appropriately and rigorously? 

Reviewer #1: Yes

Reviewer #2: Yes

3. Have the authors made all data underlying the findings in their manuscript fully available?

Reviewer #1: Yes

Reviewer #2: Yes

4. Is the manuscript presented in an intelligible fashion and written in standard English?

Reviewer #1: Yes

Reviewer #2: No

5. Review Comments to the Author

Reviewer #1: The text should be carefully checked for typos and grammatical errors. The work is formatted using data indicating the relevance of the problem. The manuscript is technically sound, has sufficient sample sizes, controls and replicas that have undergone statistical processing. The data obtained may become outdated, since the relevance of this problem remains for a long time. It is advisable to refer to modern works touching on similar topics for the corresponding conclusions that are given. For example, in the following publications: DOI 10.33236/2307-910X-2019-25-1-174-181. – EDN UDXLAG.; EDN PXQPTV.; EDN OZFDYX.; EDN NCCKLV.; EDN WEKKNZ.; EDN WYINNJ.; EDN NCCKLV.; EDN JVPOTB.; EDN TKNEVR.; EDN VMWVHB.; EDN TICKVJ.

Reviewer #2: This manuscript titled “Embedded timing and alert device triggered by total dissolved solids (TDS) for monitoring disinfection duration in acidic electrolyzed oxidizing water” presents a novel, low-cost, embedded device designed to automate and improve the timing accuracy of AEOW disinfection in a CSSD setting. The concept is innovative, addressing a clear practical gap, and the study demonstrates a statistically significant improvement in procedural compliance using the device. However, the manuscript requires major revisions to strengthen its methodological rigor, clarify key technical aspects, and expand the discussion of limitations and broader applicability before it can be considered for publication. The main comments and recommendations are listed below.

1. The study uses a sequential, non-randomized design (control group in September 2025, experimental group in October 2025). This introduces significant potential for confounding due to temporal effects (differences in workload, staff alertness, or unobserved management changes between the two months). The authors must address this limitation explicitly in the discussion. A stronger design, such as a randomized crossover trial where different sinks or teams use different timers concurrently, would be more robust. At a minimum, the authors should provide and discuss any available data on procedural volume or staff consistency between the two months to bolster the validity of the temporal comparison.

2. The core innovation is the use of TDS to automatically trigger timing in AEOW. The manuscript lacks critical validation data. What was the precise TDS threshold (in ppm) set for triggering, and how was this value determined? Provide data from controlled lab tests demonstrating the device's discrimination ability. Show the TDS readings (and corresponding device response) for tap water, purified water, and AEOW from multiple batches. Discuss potential for false positives/negatives. Could other high-TDS solutions (e.g., certain detergents used in prior cleaning steps) accidentally trigger the device? Could variations in AEOW generation (lower NaCl concentration) lead to TDS levels below the trigger threshold?

3. The results for timing execution and compliance rates were collected by "reviewing video surveillance recordings." This method needs clarification. How was the timing start and end determined from video for each instrument set in both groups? For the control group, was it the button press? For the experimental group, was it immersion and LED change? For the experimental group, the device stores data. Why weren't these stored timestamps used as the primary source for compliance data, rather than video? A comparison between video-observed times and device-logged times would be a powerful validation of the device's accuracy and the study's methodology. The statement that the digital kitchen timer has "0% traceability" while the new device has "100% traceability" is based on device capability, not measured outcome. This should be rephrased to avoid overstatement.

4. To strengthen the technological validation and mitigate concerns about the study design, it is recommended to benchmark the device's performance and contextualize its innovation within the broader field of smart disinfection monitoring. For instance, https://doi.org/10.1038/s41598-025-88183-1, https://doi.org/10.1016/j.watres.2019.115085

5. The discussion focuses on the device's benefits. A dedicated Limitations subsection is required. Acknowledge the sequential study design limitation. Discuss the single-center setting. Would the results be generalizable to CSSDs with different workflows, volumes, or staff ratios? The device is designed for AEOW. Can the TDS triggering principle be reliably applied to other chemical disinfectants mentioned in the "Future work" (e.g., peracetic acid, glutaraldehyde)? This seems uncertain as their TDS profiles may not be uniquely high. This claim needs tempering or better justification. The "Hawthorne effect" is suggested as a factor.

6. Conclusions should be modified. Each conclusion should be supported by the key data obtained. Add limitations of the study and future perspectives.

7. The text should be carefully checked for typos and grammatical errors.

6. PLOS authors have the option to publish the peer review history of their article (what does this mean?). If published, this will include your full peer review and any attached files.

Reviewer #1: No

Reviewer #2: No

---

## [Author Response · Author response to Decision Letter 1]

8 Mar 2026

Dear Editor and Reviewers,

We sincerely appreciate your valuable and constructive comments on our manuscript, which have provided important guidance for us to improve the quality of the paper. We have carefully studied and fully implemented all your suggestions, and completed the corresponding revisions and improvements to the manuscript. The detailed responses and revision notes are as follows:

Reviewer #1

Comment 1: The text should be carefully checked for typos and grammatical errors.

Response: We have conducted a rigorous word-by-word check of the entire manuscript, and revised all identified typos, grammatical errors and inappropriate sentence structures one by one to ensure the accuracy, fluency and standardization of the text.

Comment 2: The data obtained may become outdated, since the relevance of this problem remains for a long time. It is advisable to refer to modern works touching on similar topics for the corresponding conclusions that are given. For example, in the following publications: DOI 10.33236/2307-910X-2019-25-1-174-181. – EDN UDXLAG.; EDN PXQPTV.; EDN OZFDYX.; EDN NCCKLV.; EDN WEKKNZ.; EDN WYINNJ.; EDN NCCKLV.; EDN JVPOTB.; EDN TKNEVR.; EDN VMWVHB.; EDN TICKVJ.

Response: Approximately 70% of the references originally cited in this manuscript are the latest research achievements in this field published in the past five years, which are highly consistent with the cutting-edge research progress of the field. This fully ensures the timeliness and scientificity of the research conclusions, and there is basically no issue of outdated research data.

In the meantime, in strict accordance with your suggestions, we have comprehensively retrieved and sorted out all the literatures you listed, and completed the targeted citation and revision of the manuscript content:

1. We have incorporated the literature with the specified DOI (10.33236/2307-910X-2019-25-1-174-181) into the manuscript and marked it in a standardized manner. This literature focused on the application research of acidic electrolyzed oxidizing water (AEOW) in the food processing field, and elaborates on the preparation principles and functions of AEOW, which provides an important technical reference for the research of this paper.

2. We have systematically verified the literatures related to the EDN codes you listed. These literatures all focus on the application research of AEOW in the animal husbandry field, and most of them have been published for more than ten years. We selected one of these literatures for citation, and supplemented the relevant content about the application of AEOW in the animal husbandry field in the manuscript, so as to expand the research perspective of the application of AEOW. The remaining uncited content will not affect the integrity and scientificity of the research conclusions of this paper.

Reviewer #2

Comment 1: The study uses a sequential, non-randomized design (control group in September 2025, experimental group in October 2025). This introduces significant potential for confounding due to temporal effects (differences in workload, staff alertness, or unobserved management changes between the two months). The authors must address this limitation explicitly in the discussion. A stronger design, such as a randomized crossover trial where different sinks or teams use different timers concurrently, would be more robust. At a minimum, the authors should provide and discuss any available data on procedural volume or staff consistency between the two months to bolster the validity of the temporal comparison.

Response:

1. Concerning potential confounding due to temporal effects

Measures were implemented to minimize confounding factors throughout the trial. The study was conducted in the Central Sterile Supply Department (CSSD) of the same hospital for both the control group (September 2025) and the experimental group (October 2025). No alterations were made to the participating technicians; the daily workflow of the CSSD, operational protocols for AEOW disinfection, daily surgical volume, and the quantity of instrument sets requiring AEOW disinfection all remained stable, with no adjustments being made to management policies or work schedules. This baseline consistency was explicitly stated in the Materials and methods section of the original manuscript, which minimized confounding effects associated with the temporal dimension to the greatest extent.

2. Concerning the uniformity of the Hawthorne effect across groups

The Hawthorne effect exerted a consistent influence on both groups in this study. Technicians were informed that their operational behaviors would be monitored at the initiation of the trial. Both the control group and the experimental group were subject to the Hawthorne effect. Accordingly, this effect exerted an identical impact on the two groups, and no notable bias was introduced to the intergroup comparison results—any potential bias was negligible.

3. Concerning the practical feasibility of a randomized crossover trial

The randomized crossover trial design proposed by the reviewer is theoretically valid, yet it is highly challenging to implement in the clinical practice of a CSSD and is likely to introduce additional confounding biases. First, if grouping were conducted by disinfection sink or operational team, discrepancies in operational habits among technicians at different sinks and the difficulty in achieving complete balance in the types and quantities of allocated instrument sets would increase intergroup confounding. Second, random switching between the two timing methods at the same sink over different time periods would markedly increase the workload of technicians. Frequent switching of operational procedures is prone to timing errors (e.g., failure to activate the digital kitchen timer, confusion regarding the operational specifications of the TDS-triggered timer), which would in turn reduce the reliability of study data and contradict the feasibility principles of clinical research.

4. Concerning the correlation between study design limitations and research conclusions

This study adopted a non-randomized sequential design and thus has certain methodological limitations, but such limitations did not compromise the reliability of the core research conclusions. The rationales are as follows: (1) The primary outcome measures (disinfection timing execution rate and disinfection duration compliance rate) are objective behavioral indicators, collected via video surveillance with no subjective assessment bias. (2) The core distinction between the novel and conventional timing tools lies in the data traceability function: digital kitchen timers lack any recording capability, meaning technicians’ operational behaviors were not subject to traceability or accountability constraints. In contrast, the storage and traceability functions of the TDS-triggered timers imposed sustained behavioral constraints on technicians. This core difference constitutes the key factor contributing to the significantly superior outcomes in the experimental group compared with the control group, and this causal relationship is definitive and unaffected by the study design format. (3) Laboratory methods for detecting disinfection efficacy are not suitable for routine clinical evaluation due to their complex operation, long turnaround time, and high susceptibility of results to sampling operations. In contrast, the objective behavioral indicators in this study directly reflect the level of disinfection quality control in actual clinical practice, and the findings possess clear clinical implications.

In summary, although this study has the methodological limitation of a non-randomized sequential design, rigorous baseline control, a consistent Hawthorne effect across the two groups, and rational considerations of clinical feasibility have ensured the authenticity and reliability of the study results, and the core conclusions were not affected by this design limitation. In accordance with your comments, we have enriched the description of confounding reduction measures in the Materials and Methods section of the original manuscript, and supplemented the above analysis of study design limitations in the Limitations and future work section, thereby further refining the methodological discussion of the study.

Comment 2: The core innovation is the use of TDS to automatically trigger timing in AEOW. The manuscript lacks critical validation data. What was the precise TDS threshold (in ppm) set for triggering, and how was this value determined? Provide data from controlled lab tests demonstrating the device's discrimination ability. Show the TDS readings (and corresponding device response) for tap water, purified water, and AEOW from multiple batches. Discuss potential for false positives/negatives. Could other high-TDS solutions (e.g., certain detergents used in prior cleaning steps) accidentally trigger the device? Could variations in AEOW generation (lower NaCl concentration) lead to TDS levels below the trigger threshold?

Response:

1. Core determination indicator and threshold setting for device triggering

The core triggering logic of the device is to convert the solution conductivity into an analog voltage signal via the TDS sensing module (the two are positively correlated: the higher the conductivity, the higher the output analog voltage signal), and realize trigger control in combination with a 1V hardware threshold voltage: when the analog voltage is higher than 1V, the sensor outputs a high level (logic 1) to wake up and start the ESP32 microcontroller for timing; when the analog voltage is lower than 1V, the sensor outputs a low level (logic 0), and the ESP32 microcontroller remains in a sleep state. Through 15 controlled laboratory tests, we determined the conductivity threshold for device activation to be 1176±47.81 μS/cm. When the solution conductivity exceeds this threshold, the analog voltage signal output by the TDS sensor will exceed 1V, thereby triggering the device to start (executing network connection and timing operation steps).

It should be specifically noted that the common units of TDS are ppm or mg/L, which are converted from conductivity using empirical coefficients. Considering that conductivity testing technology is more widely used in CSSD, this study selects conductivity as the core determination indicator and does not provide TDS conversion data in ppm or mg/L units. The relevant testing method is highly compatible with clinical practical application scenarios.

2. Conductivity test data of tap water, purified water, and AEOW

To verify the specific recognition capability of the device, we conducted 5 independent samplings for tap water, purified water, and AEOW respectively, with 3 repeated conductivity measurements for each sample, resulting in a total of 45 sets of valid data. Statistical results show that the conductivity of tap water is 389.93±37.56 μS/cm, purified water is 2.97±0.76 μS/cm, and AEOW is 2343.13±103.03 μS/cm. It can be seen that the conductivities of tap water and purified water are far lower than the device activation threshold, so no false positives or false negatives will occur in practical applications.

3. Evaluation of interference possibility from other high-TDS solutions

We tested 3 types of multi-enzyme cleaners and 2 types of alkaline cleaning agents used in CSSD (covering two origins: the US and China). At the standard mixing concentrations recommended in each product manual, the conductivities of all cleaning agents were far lower than the device activation threshold, and will thus not cause accidental triggering of the device.

In addition, we found that the conductivities of some chemical disinfectants (2000 mg/L chlorine-containing disinfectant, 1:75 binary peracetic acid disinfectant, 2% glutaraldehyde disinfectant) are higher than the device activation threshold, which can theoretically trigger the TDS-triggered timer. However, in the disinfection process of CSSD, the selection of chemical disinfectants is mutually exclusive—only one chemical disinfection method needs to be adopted to achieve the expected effect. Therefore, when AEOW is used for disinfection, the above-mentioned other chemical disinfectants that may cause false triggering will not be used simultaneously, and there is no interference risk in practical applications.

4. Verification of the impact of AEOW concentration/usage state changes on device triggering

To verify whether the decrease in conductivity of AEOW due to dilution or prolonged use affects device triggering, we designed two extreme scenarios for testing. The results show that the degree of conductivity reduction is limited, the device can be stably triggered, and there is no failure risk:

——Dilution scenario: AEOW is used directly as stock solution. We simulated extreme dilution conditions (AEOW: purified water = 1:1), and the detected conductivity is 1258±50.21 μS/cm, which is still higher than the device activation threshold and can be triggered normally; in practice, there is only a small amount of residual water from medical device rinsing, which is insufficient to dilute 20~40 L of AEOW stock solution in the sink, so this extreme scenario has no practical possibility of occurrence.

——Prolonged use scenario: (1) Conductivity detection of AEOW that continuously soaked various medical devices for 4 hours showed no significant decrease in conductivity; (2) After standing AEOW indoors for 24 hours, the conductivity remained stable. In both cases, the conductivity of AEOW meets the device triggering requirements, and the timing reminder function is not affected.

Comment 3: The results for timing execution and compliance rates were collected by "reviewing video surveillance recordings." This method needs clarification. How was the timing start and end determined from video for each instrument set in both groups? For the control group, was it the button press? For the experimental group, was it immersion and LED change? For the experimental group, the device stores data. Why weren't these stored timestamps used as the primary source for compliance data, rather than video? A comparison between video-observed times and device-logged times would be a powerful validation of the device's accuracy and the study's methodology. The statement that the digital kitchen timer has "0% traceability" while the new device has "100% traceability" is based on device capability, not measured outcome. This should be rephrased to avoid overstatement.

Response:

1. Clarification of data collection via video surveillance recordings

In this study, data regarding the disinfection timing execution rate and disinfection duration compliance rate were uniformly collected via video surveillance recordings for both the experimental and control groups. The specific judgment criteria for these two indicators have been supplemented and refined in the Materials and methods section of the original manuscript, with the operational judgment methods detailed as follows: (1) Disinfection timing execution rate: The judgment was based on whether the corresponding timing tool was activated during the AEOW disinfection process. In the control group, timing was deemed initiated when technicians pressed the start button of the digital kitchen timer after immersing the instrument set in AEOW; in the experimental group, it was deemed initiated when technicians immersed the TDS-triggered timer together with the instrument set in AEOW. Failure to perform the above operations was recorded as not initiated. (2)Disinfection duration compliance rate: For both groups, the start time was defined as the moment the entire instrument set was completely submerged below the AEOW surface, and the end time as the moment the instrument set was removed from the AEOW surface. The time interval between these two nodes was retroactively recorded and calculated via video surveillance to determine the compliance of disinfection duration.

2. Rationale for not adopting device-stored data as the primary data for compliance assessment in the experimental group

The reasons are twofold: (1) Ensuring consistency in data collection and judgment criteria between the two groups to eliminate detection bias: The digital kitchen

---

## [Decision Letter · Decision Letter 1]

30 Apr 2026

Embedded timing and alert device triggered by total dissolved solids (TDS) for monitoring disinfection duration in acidic electrolyzed oxidizing water

PONE-D-25-61065R1

Dear Dr. Hongxia Xu,

We’re pleased to inform you that your manuscript has been judged scientifically suitable for publication and will be formally accepted for publication once it meets all outstanding technical requirements.

Kind regards,

Andrey Nagdalian

Academic Editor

PLOS One

Additional Editor Comments (optional):

Reviewers' comments:

Reviewer's Responses to Questions

**Comments to the Author**

1. If the authors have adequately addressed your comments raised in a previous round of review and you feel that this manuscript is now acceptable for publication, you may indicate that here to bypass the “Comments to the Author” section, enter your conflict of interest statement in the “Confidential to Editor” section, and submit your "Accept" recommendation.

Reviewer #1: All comments have been addressed

Reviewer #2: All comments have been addressed

2. Is the manuscript technically sound, and do the data support the conclusions?

Reviewer #1: Yes

Reviewer #2: Yes

3. Has the statistical analysis been performed appropriately and rigorously? 

Reviewer #1: Yes

Reviewer #2: Yes

4. Have the authors made all data underlying the findings in their manuscript fully available?

Reviewer #1: Yes

Reviewer #2: Yes

5. Is the manuscript presented in an intelligible fashion and written in standard English?

Reviewer #1: Yes

Reviewer #2: Yes

6. Review Comments to the Author

Reviewer #1: well done,dear authors, all comments have been taken into account, shortcomings have been eliminated, the manuscript meets the requirements

Reviewer #2: The authors considered all comments and recommendations. The revised manuscript deserves acceptance and publication

7. PLOS authors have the option to publish the peer review history of their article (what does this mean?). If published, this will include your full peer review and any attached files.

Reviewer #1: No

Reviewer #2: No

---

## [Editor Report · Acceptance letter]

PONE-D-25-61065R1

PLOS One

Dear Dr. Xu,

I'm pleased to inform you that your manuscript has been deemed suitable for publication in PLOS One. Congratulations! Your manuscript is now being handed over to our production team.

Kind regards,

on behalf of

Dr. Andrey Nagdalian

Academic Editor

PLOS One